


# On the Relationship Between Flood and Contributing Area

Spence Christopher[1] and Samson.Girma Mengistu[2]

[1]Environment and Climate Change Canada, Saskatoon, SK, CANADA
5 [2]University of Alberta, Edmonton, AB, CANADA

*Correspondence to*: Christopher Spence (chris.spence@canada.ca)





**Abstract:** While it is well known that the vast majority of the time only a portion of any watershed contributes runoff to the outlet, this extent is rarely documented. The power-law form of the streamflow and contributing area (Q-$A_c$) relationship has been known for a half century, but it is uncommon for it to be quantified or its controls evaluated. In this study a semi-

distributed hydrological model (MESH-PDMROF) that can simulate contributing area and streamflow was employed to compare contributing area and flood frequency distributions in a southern Manitoba, Canada catchment and test the hypothesis that the relationship between a catchment's floods and contributing area is a power function that influences the form of regional flood-area relationships. The model simulated streamflow reasonably well (Nash Sutcliffe

values = 0.62). Modelled estimates of the area contributing to the mean annual flood were much lower (0.3) than those derived from independent topographic analysis (0.9) described in earlier literature, even after bias and error corrections. Estimates of the coefficient and exponent of the Q-$A_c$ power law function ranged from 0.08 – 0.14 and 0.9 - 1.12, respectively. Lower exponent values of regional flood frequency curves suggest they are a construct of Q-$A_c$ curves from

individual basins. The non-linear nature of this relationship implies any contributing area change will have a profound impact on flood magnitude. The mean annual flood of the major river in this region, the Red, has increased 33% since 1987. Applying the coefficient and exponent ranges above suggests this is associated with an expansion in contributing areas of 29 – 38%. There are implications for the attribution of causes and mitigation of nutrient transport from

regional watersheds. However, how physiography and land and water management could change Q-$A_c$ power law exponents is poorly known and MESH-PDMROF does not provide explicit estimates of the spatial distribution of contributing area. These are areas encouraged for future research.



## 1 Introduction

The concept that the dynamics of runoff contributing area control streamflow yield and flood

magnitude has existed for at least half a century. Betson (1964) was among the first to suggest

the idea that only a portion of a catchment supplies water to the outlet when he introduced the

partial area concept which describes the spatial manifestation of Hortonian overland flow. After

saturation overland flow was identified (Hewlett and Hibbert, 1963), the perception of possible

contributing area controls and dynamics expanded as it became recognized that subsurface

conditions could also control expansion and contraction of these areas (Hewlett and Hibbert,

1967; Dunne and Black, 1970). If 'contributing area' can be defined as that area that provides

water to a catchment outlet over a defined period, for instance a rainfall-runoff event and a

water-year, it has become evident from the literature (Beven and Wood, 1983; Devito et al.,

2005; Tetzlaff et al., 2007) that how this area manifests is not only a function of predominant

hydrological processes, as noted above, but also heterogeneity in landscape topography,

topology and typology.

Static physiographic characteristics control where contributing area is likely to occur, but it is

variable atmospheric and soil climate conditions that control the temporal variability of

contributing area. The state of contributing area is a function of the magnitude of water

available, the distribution of water storage, and rates of loss along runoff pathways. Some of the

original investigations of the relationship between contributing area and runoff response were

based upon the idea that the fraction of the basin that was an effective contributing area during a

storm was simply the ratio of storm runoff to effective precipitation (Dickinson and Whiteley,

1970) and this approach has been applied to evaluate relationships between contributing area and





antecedent conditions (Gburek, 1990). Dry regions or periods typically experience conditions where there is no area contributing to flow, essentially resulting in the disappearance of the stream. The distribution and application of water to the catchment during such conditions can have a profound impact on the extent of contributing areas and the duration they remain, and in

turn, how catchment streamflow responds (Jencso et al., 2009; Spence et al., 2010). It would be wrong to suggest contributing area dynamics are arguably more predictable in wetter conditions because saturated portions of the landscape have more persistence and runoff pathways are engaged with the stream along a smooth continuum (Dunne, 1978). Evidence suggests there remain significant non-linearities in the relationship between contributing area and runoff

response (Dickinson and Whiteley, 1970; James and Roulet, 2007; Ali and Roy, 2010).

This issue of importance to hydrology is significant for biogeochemistry. The influence of contributing area behaviour is implicitly understood to be very important for solute fluxes. The concepts of 'hot moments' and 'hot spots' (McClain et al., 2003; Bernhardt et al., 2017) capture

this idea that there are locations and periods that provide disproportionate sources of chemical loads to streams and lakes. To explain and solve many of today's problems associated with contaminants and excess nutrients in the aquatic ecosystem and human water supply, it is not only necessary to identify the extent and location of contributing areas of solutes, but also the frequency and duration with which these areas are engaged. The contributing area frequency

distribution dictates the characteristics of the periods during which constituents remain to be processed on the landscape, as well as their rate of flushing (Creed et al., 1996), which is important for controlling chemical concentrations, and in turn, loads.




Unfortunately, beyond the methods discussed above that assume the runoff ratio is a substitute for contributing area fraction, there are few examples of contributing area measurement techniques that will enable analysis of how contributing area dynamics are controlled by climate or landscape traits.   There are common topographic index methods to estimate contributing area

(Beven and Wood, 1983), but these assume saturation overland flow and the variable source area concepts are applicable. Mapping would be very useful for model validation, but is uncommon, generally applying either remote sensing (Phillips et al., 2011) or field observations (Spence et al., 2010).  Other methodologies include the use of soil moisture indices (James and Roulet, 2007), aquatic chemistry (Ali et al., 2010), or those that map stream networks (Godsey and

Kirchner, 2014) from which contributing area can be deduced.  Few of these studies have developed time series robust enough to elaborate on the frequency distribution of contributing areas.  Jencso et al. (2009), Smith et al., (2013) and Reaney et al. (2013) are excellent examples of how numerical models have been applied to fill this gap.  By explicitly accounting for hydrological connections, these models have been used to identify the frequency, duration or

extent of contributing areas in actual and synthetic watersheds.  These studies were able to describe some key aspects of how contributing area traits are related to flood magnitude, but did not determine if the fundamental contributing area – streamflow relationship follows a power law function of the form first hypothesized by Gray (1961):

$$Q = a \cdot A^b \tag{1}$$

Before Jencso et al., (2009), the water resource community had not traditionally conceptualized contributing area as having traits similar to floods, such as frequency, duration and extent. Perhaps the closest useful example is from Canada.  Contributing areas expected to produce the



mean annual flood have been estimated for numerous Canadian Prairie watersheds.  Many parts

of the post-glacial landscape of the Canadian Prairies have very poorly defined drainage

networks because of a semi-arid climate combined with hummocky or very flat terrain.  Large

variation in drainage area was documented over 50 years ago (Stichling and Blackwell, 1957),

with the consequence of high uncertainty in some catchment boundaries.  Agriculture Canada

(1983) following concepts and methods outlined in Stichling and Blackwell (1957) defined two

specific contributing area states that have been applied for decades within Canada, but have

universal application.  First, the gross drainage area to a stream ($A_g$) at a specific location is the

plane area enclosed by its drainage divide that might be expected to entirely contribute runoff to

a specific location under extremely wet conditions.  The gross drainage area is typically

estimated by surface topographic divides and is what most hydrologists would define as the

catchment area.  Second, the effective drainage area ($A_e$) is that portion of $A_g$ that might be

expected to entirely contribute runoff to that location during the mean annual flood.  Any

difference between $A_g$ and $A_e$ is typically caused by natural areas of high storage capacity (e.g.,

wetlands or lakes, but may include upper reaches of any catchment) that prevent runoff from

reaching the catchment outlet in a year of mean runoff.   In contrast to $A_g$, which in theory is a

constrained by topography, $A_e$ is delineated with a conceptual line that encapsulates the area

producing the mean annual flood.   In some watersheds, runoff is intercepted by major

depressions from which there is no drainage even under extremely wet conditions.  Agriculture

Canada (1983) refers to the area contributing runoff to such depressions as dead drainage areas,

but more common terms are "internally-drained" or endorhic watersheds.





These estimates of effective drainage area are broadly accepted by the Canadian water resource community for use in water management applications. This same community, however, recognizes that these estimates have never been tested with field measurements during a mean annual flood. Furthermore, while it is well known that contributing areas fluctuate spatially and

temporally with meteorological inputs, and this has profound influence on flood magnitude (Ehsanzadeh et al., 2012; Kusumastuti et al., 2008) there are few examples of estimates of contributing area frequency distributions (e.g., Jencso et al., 2009) and none for this region. The models of Smith et al. (2013) and Reaney et al. (2013) are valuable for estimating contributing area over the required timescales, but they are not necessarily suited to larger watersheds. At this

scale it may be more appropriate to apply semi-distributed numerical hydrological models. In this study, the objective was to estimate the frequency distribution of the contributing area of a meso-scale catchment (~2000 km$^2$). This was done with a semi-distributed numerical hydrological model scheme that has been proven to adequately estimate contributing area (Mengistu and Spence, 2016). This present study builds on that work and has the goals of 1)

comparing contributing area and flood frequency distributions and; 2) tesing the hypothesis that the relationship between a catchment's floods and contributing area is a power function that influences the form of regional flood-area relationships.

## 2   Study Basin

The La Salle River Watershed (Figure 1) encompasses 2400 km$^2$ in southeastern Manitoba, Canada. Elevation in the watershed ranges between 329 and 226 metres above sea level, and the watershed drains east to the Red River and eventually Lake Winnipeg. Climate within the La Salle is semi-humid in nature with mean annual precipitation and temperature of 560 mm and



2.5°C, respectively. Mean summer (JJA) and winter (DJF) temperatures are 16.5°C and -13.0°C respectively. Almost 78 percent of the average annual precipitation in the watershed falls in the form of rain while the rest appears as snow. The primary surficial geology of the watershed is glaciolacustrine sand and clay but alluvial sediments are common in the northwestern corner of

the watershed. Deposits vary in thickness from 5m in the east to 50m in the northwest. Clay soils predominate covering 74% of the watershed with the remaining 26% typically of sand/silt-loam texture (Leon et al. 2010). In the past, the area was dominated by extensive grassland and several large wetlands (Bossenmaier and Vogel, 1974). However, agriculture related drainage has removed 99% of wetlands, causing it to be one of the most severely drained landscapes in North

America (Melles et al. 2010). Today, the watershed is overwhelmingly under annual cereal and oil seeds production.

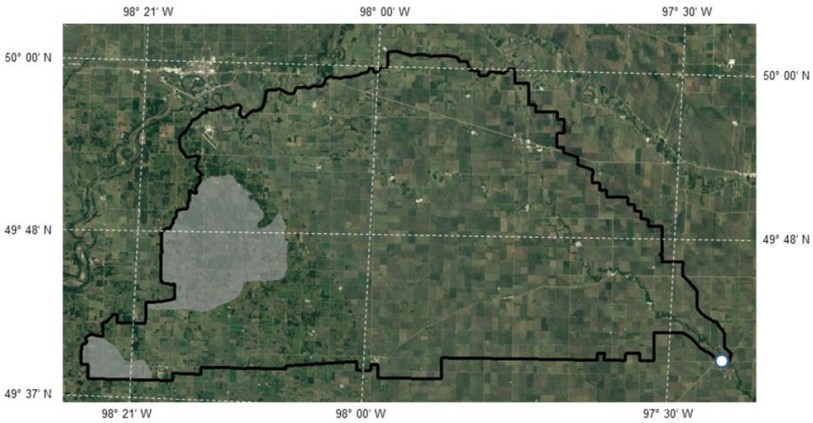

Figure 1:  The La Salle River Basin outlined in black. The white dot denotes the location of the Water Survey of Canada hydrometric gauge 05OG001.  The grayed areas are those identified as
non-contributing during the mean annual flood by Agriculture Canada (1983).



## 3 Methodology

### 3.1 MESH Model Description

In the late 1990's, the Canadian atmospheric and hydrologic science community realized that to answer many of the most pressing societal questions being posed, the coupling of atmospheric

and hydrological process present in nature would need to be incorporated into model structures. Portions of the Canadian Land Surface Scheme (CLASS; Verseghy, 1991; Verseghy et al., 1993) and the distributed hydrological routing scheme from WATFLOOD (Kouwen et al., 1993) were merged and eventually become one of Environment and Climate Change Canada's land surface schemes; MESH (Modélisation Environmentale Communautaire (MEC) - Surface and

Hydrology). The algorithms selected from CLASS simulate vertical energy and water budgets of soil, snow, and vegetation (Maclean et al., 2010). The WATFLOOD model contributed lateral routing algorithms (e.g., WATROF) (Soulis et al., 2000), stream routing algorithms employing continuity equations and Manning's formula, and the sub-grid conceptual grouped response unit approach (GRU) (Kouwen et al., 1993), which is an assumed homogenous, but perhaps not

contiguous, unit of the watershed. MESH operates on a daily time step or less. Pietroniro et al. (2007) summarizes the MESH design.

MESH simulates runoff at any resolved point within a catchment. The coupled model laterally routes excess surface runoff (that above a defined ponding depth) and subsurface runoff (water in

excess of soil storage) from the vertical water budget through soils and stream channels to watershed outlets. The vertical water budget algorithms are run on each grouped response unit independently and the weighted area of each GRU is used to calculate the overall fluxes, which are then processed with the lateral routing algorithms. The current version of MESH uses





CLASS 3.6. MESH and its immediate predecessor, WATCLASS, have been successfully applied

to several Canadian basins (Pohl et al., 2005; Davison et al., 2006; Pietroniro et al., 2007; Soulis

and Seglenieks, 2007; Dornes et al., 2008; Yirdaw et al., 2009; MacLean et al., 2010).

The introduction of a new lateral surface runoff transfer algorithm (Probability Distribution

Model based RunOFf generation, PDMROF) (Mekonnen et al., 2014) is a new alternative to

WATROF.  The assumptions of the grouped response unit approach are violated in most

catchments (Beven and Wood, 1983; Devito et al., 2005; James and Roulet, 2007; Jencso et al.,

2009) and PDMROF represents an attempt to represent the influence of heterogeneous storage

deficits on the variable nature of runoff contributing areas. PDMROF is conceptually similar to

PDM (Moore, 2007), which assumes point generated runoff (e.g., catchment, sub-catchment or

grid cell) is a function of the storage capacity of the soil at that point. PDMROF divides grouped

response units into distinct units with different soil storage capacities and employs a Pareto

probability distribution to represent spatial variation of soil storage capacity across the

watershed. Runoff at any point is soil moisture excess which is integrated with the Pareto

distribution to calculate total runoff. PDMROF improves streamflow predictions as compared to

the static contributing area assumptions of WATROF (Mekonnen et al., 2014) and can

adequately simulate runoff contributing area fraction (Mengistu and Spence, 2016).

*3.2 MESH Modeling Data and Data Preprocessing*

*3.2.1 Meteorological data*

Hourly precipitation data for the period 2002 to 2014 were obtained from the 5 km resolution

Canadian Precipitation Analysis (CaPA), (Mahfouf et al., 2007) while the remaining forcing





inputs were derived from the regional configuration of Environment and Climate Change

Canada's Global Environmental Multi-scale (GEM) Numerical Weather Prediction (NWP)

model (Mailhot et al., 2006). The CaPA uses surface synoptic reports of 6 hour precipitation

from Environment and Climate Change Canada's historical weather and climate archives and

combines it with archived 6 hour precipitation forecasts from the 15 km GEM grid as a

background field to account for orographic effects, discontinuities in precipitation measurement,

poor rain gauge density and snowfall measurement uncertainty. The GEM background field is

considered to be the best option for estimating snowfall, particularly in regions where snowfall

observation can be unreliable (Mekonnen et al., 2014), like southern Manitoba.  These data have

proved to work well in similar applications to the east in Saskatchewan (Mengistu and Spence,

2016).

### 3.2.2 Hydrometric data

MESH calibration and validation requires daily streamflow observations. The Water Survey of

Canada operates a hydrometric station on the La Salle near Sanford (05OG001) above its

confluence with the Red River.  Daily streamflow estimates from this station were obtained from

http://www.ec.gc.ca/rhc-wsc/ for the period 2002-2014 to coincide with the period of record

from CaPA.

### 3.2.3 Topography and land cover data

Elevation data for the La Salle watershed were obtained from the CGIAR-CSI SRTM 90m DEM

Digital Elevation Database v 4.1 (http://srtm.csi.cgiar.org/SELECTION/inputCoord.asp). A 30m

resolution digital elevation model was re-sampled from the 90m SRTM data using bilinear





interpolation. Land use data were obtained from the 2001 Manitoba Land Initiative's (MLI) land

cover database (https://mli2.gov.mb.ca/). The original fourteen land cover classifications were

merged into eight classes of agriculture, deciduous forest, water, grassland, wetland, urban,

mixed forest and coniferous forest. Soil and vegetation parameters were obtained from the

Canadian Soil Information System (CanSIS, http://sis2.agr.gc.ca/cansis).  A model grid

resolution of 10 km was selected within which was one GRU for each of the eight land cover

types.  The gross drainage area boundary of the La Salle River above Sanford and its drainage

networks, slopes and channel lengths were delineated using Green Kenue (Ensim Hydrologic,

2007). The parameterization capability of Green Kenue was employed to populate land cover

distribution for each GRU.

### 3.3 MESH Model Set Up and Model Optimization

Spin up of the model used calendar year 2001 while the remaining periods 2002 - 2009 and 2010

– 2014 were used for calibration and validation, respectively. Calibration parameters were

adjusted using OSTRICH (Optimization Software Toolkit for Research Involving Computational

Heuristics; Matott, 2005) until acceptable model performance statistics were achieved. This

employed 10000 model realizations applying Monte Carlo sampling of different combinations of

model calibration parameters from within sampling ranges listed in Table 1.

Those combinations of parameters that resulted in the best simulated streamflow at the watershed

outlet were selected as optimal parameter values (Table 1). The best simulated streamflow was

that which had the highest Nash-Sutcliffe Efficiency (NSE) coefficient when compared against

observed streamflow. Values of NSE range between -∞ and 1.  When NSE < 0 the arithmetic

average of the observations is a better predictor than the model; when NSE = 0 the model has the



same predictive power as the arithmetic average of the observations; and when NSE > 0 the

model is a better predictor with prediction accuracy increasing towards 1, which represents

complete agreement between observed and simulated flows (Nash and Sutcliffe, 1970).

Table 1:  Most influential MESH-PDMROF parameters on streamflow simulation.

| Calibrated parameter | Description | Calibration range | Calibrated Value |
|---|---|---|---|
| SDEPROW | Permeable depth of the soil column | 0 – 5 | 0.50 |
| ZSNLROW | Limiting snow depth below which coverage is <100% | 0 - 5 | 0.14 |
| CMINROW | Minimum storage capacity parameter for the Pareto distribution function | 0 - 20 | 0.00 |
| CMAXROW | Maximum storage capacity parameter for the Pareto distribution function | 0 - 20 | 3.34 |
| K1ROW | Time constant for the first linear reservoir for the Pareto reservoir function | 0 - 60 | 59.91 |
| K2ROW | Time constant for the second linear reservoir for the Pareto distribution function | 0 - 60 | 19.98 |
| WF_R2 | River roughness | 0.001 – 2.0 | 0.015 |
| BCROW | Shape factor parameter for the Pareto distribution function | 0 - 10 | 3.27 |
| SAND1 | Percent content of sand in the mineral soil (Layer 1) | 0 - 20 | 7.66 |
| CLAY1 | Percent content of clay in the mineral soil (Layer 1) | 0 - 80 | 64.6 |
| SAND2 | Percent content of sand in the mineral soil (Layer 2) | 0 - 15 | 12.4 |
| CLAY2 | Percent content of clay in the mineral soil (Layer 2) | 0 - 85 | 45.0 |
| SAND3 | Percent content of sand in the mineral soil (Layer 3) | 0 - 20 | 9.0 |
| CLAY3 | Percent content of clay in the mineral soil (Layer 3) | 0 - 80 | 72.3 |

## 3.4 Estimation of Runoff Contributing Area

The gross and effective drainage areas delineation process applied in the agricultural zone of the

Canadian Prairie Provinces is described in Agriculture Canada (1983) and briefly summarized

10     here.  Water Survey of Canada gauges were demarcated on National Topographic System


1:50,000 topographic maps and drainage divides defining $A_g$ to these points hand drawn

perpendicular to contour lines that defined the height of land.  Several factors were considered

when defining $A_e$, including the number and size of depressions relative to the upslope area,

topographic slope.  Aerial photographs, field inspection and interviews with local residents were

used to augment information from the topographic maps.  Two technicians delineated areas

independently, which were then compared and discrepancies were resolved by mutual

satisfaction.  Areas were measured with either an electronic digital read-out planimeter or an

electronic D-Mac digitizer, which output boundary coordinates via a keypunch machine to

computer cards which were input to an in-house Water Survey of Canada basin calculation

program.  In the 1990's, $A_g$ and $A_e$ boundaries from the original topographic maps were re-

digitized into a geographic information system (GIS) (Agriculture and Agri-Food Canada, 2001).

It is the areas from this GIS database that were used in the current study.  Gross and effective

drainage areas of the La Salle River watershed near Sanford are estimated to be 1825 km$^2$ and

1645 km$^2$, respectively, providing an $A_e$ fraction ($A_{ef}$) of 0.9.  Following methods from Mengistu

and Spence (2016), half hourly modelled estimates of runoff contributing area to the watershed

outlet were output from each grid and subsequently averaged to compute a daily value for the

grid. The total daily contributing area was obtained by summing each grid estimate from each

day, and the daily contributing area fraction, $A_{cf}$, was calculated by dividing this value by $A_g$.

*3.5 Frequency Analysis*

Annual maximum contributing area and observed and simulated annual maximum streamflow

coincident time series (2002-2014) were tested for their goodness of fit to both the Generalized





Extreme Value and Log Pearson Type III distributions using methods from Laio (2004).

Methods employed the Darling-Anderson test statistic ($A^2$):

$$A^2 = -n - \frac{1}{n}\sum_{i=1}^{n}[(2i-1)\ln[F(x_i,\theta)] + (2n+1-2i)\ln[1-F(x_i,\theta)]] \qquad (2)$$

where n is the sample size, and $F(x_i,\theta)$ is the cumulative distribution function of either

distribution. The coincident streamflow and contributing area time series were short and may not

represent the real population from which they come. For instance, neither contributing areas nor

streamflow were expected to have reached their maximum values during the study period.

Furthermore, literature implies that underlying mechanisms behind contributing area dynamics

for different return periods may vary and be described best as a mixture of distributions

(Ehsanzadeh et al., 2012). If the samples of contributing area or streamflow did not fit either the

Generalized Extreme Value or Log Pearson Type III distributions, plotting position was used to

estimate return period, T(x). The probability that a contributing area (or streamflow value) larger

than *x* will occur in any given year is:

$$P(x) = 1 - F(x) \qquad (3)$$

where F(x) is the probability that a value larger than *x* will not occur and T(x) is the inverse of

P(x):

$$T(x) = 1/P(x) = 1/(1 - F(x)) \qquad (4)$$

The rank value (m) of the ascending ordered values of x was used to determine an artificial

plotting position, F(m), for that value of x.  A single simple distribution free plotting position

may best be described with (Cunnane, 1978; Guo, 1990):

$$F(m) = (m - {}^2\!/_5)/(n + {}^1\!/_5) \qquad (5)$$





where n is sample size. Contributing area fraction ($A_{cf}$) and maximum annual streamflow ($Q_{max}$) frequency curves were constructed using either these unbiased plotting positions or those from the underlying probability distribution.

## 4 Results

### 4.1 Streamflow

MESH-PDMROF produced reasonable simulations of streamflow during the eight year calibration and five year validation periods. The model was able to reasonably capture daily variation in streamflow (Figure 2). Model performance statistics revealed an acceptable calibration period NSE value of 0.62 with minimal decrease to 0.59 during the validation period. The model captured very well the timing of peak flow in most years, and low or zero flow conditions in all years. It is encouraging that the model also captured the timing of flow cessation in most years, which is very important in intermittent streams such as the La Salle. However, the model underestimated the magnitude of peak annual streamflow in some years (e.g., 2004, 2010 and 2013).

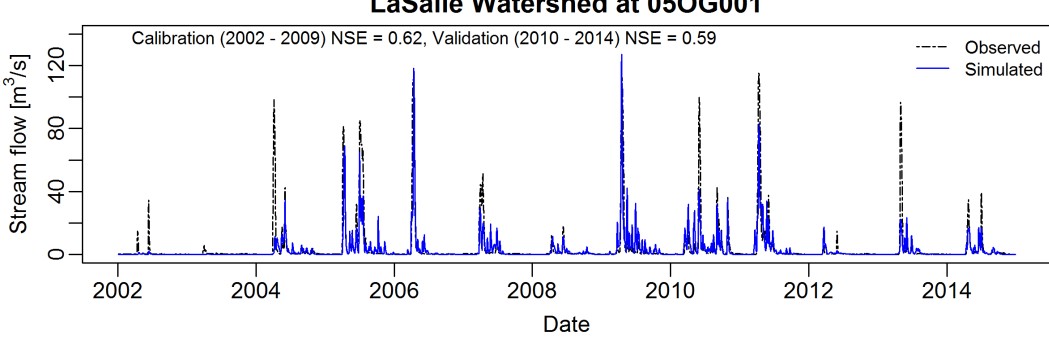

Figure 2: Daily time-series graph of MESH-PDMROF simulated versus observed streamflow (2002-2014) at the WSC gauge 05OG001.


The mean annual flood estimated from observed and simulated data was 60 and 30 m$^3$/s,

respectively. Error in the estimate of the 1:20 year return period event was 20% with observed

and simulated estimates of maximum annual streamflow of 168 m$^3$/s and 124 m$^3$/s, respectively.

This difference was typical with an overall negative bias in simulated $Q_{max}$ of 18%. This

5  underestimation of peak annual streamflow is reflected in the flood frequency distributions

(Figure 3). Differences in the first four moments of the coincident observed and simulated time

series (Table 2) influenced the ability of MESH-PDMROF to represent the Log Pearson Type III

distribution exhibited by the observed $Q_{max}$ time series ($A^2$=0.57 p=0.01). The Anderson Darling

statistics of the simulated $Q_{max}$ time series for the Log Pearson Type III and the GEV

10  distributions were 0.21 (p=0.99) and 0.23 (p=0.31), respectively, suggesting neither distribution

was appropriate, so plotting positions were determined using Eqs.4 and 5 and are illustrated in

Figure 3.

Table 2: Moments of the Observed and Simulated $Q_{max}$ times series.

|  | Mean | Standard Deviation | Skewness | Kurtosis |
|---|---|---|---|---|
| Observed | 68 | 42 | -0.3 | -1.7 |
| Simulated | 44 | 42 | 1.0 | -0.1 |





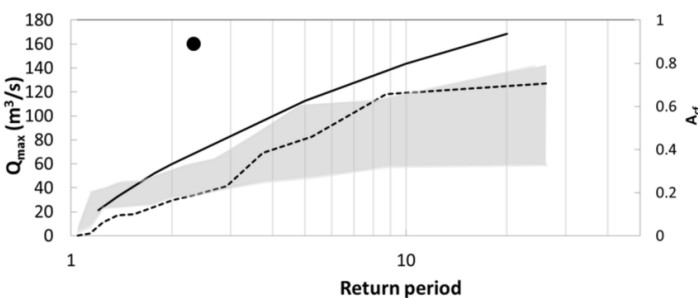

Figure 3: Flood frequency curves derived from the period 2002-2014 of observed (solid black line) and simulated (dashed black line) La Salle River annual maximum streamflow. The grey area denotes the probable range of contributing area fraction for each return period. The black dot represents the estimate of effective drainage area following methods by Agriculture Canada (1983).

## 4.2 Contributing Area

The temporal synchrony of simulated contributing area and observed streamflow (Figure 4) shows high (low) streamflow peaks were coincident with large (small) contributing areas. Sometimes the rising limb of the hydrograph was associated with abrupt increases in contributing area (e.g., 2009), indicating the sensitivity of modelled streamflow to expansion in contributing area. Following annual peak contributing area, the rate of streamflow recession was often faster than the rate of contributing area contraction. In some cases, such as 2004 and 2014, the maximum streamflow was not associated with maximum contributing area. Flow peaks in these two years coincided with the second largest simulated contributing area. Zero flow was almost always associated with no contributing area.





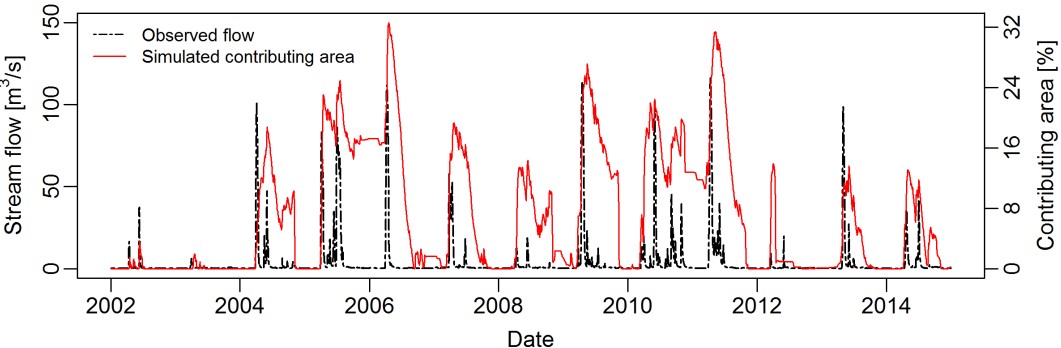

Figure 4: 2002-2014 time series of MESH-PDMROF simulated contributing area and observed streamflow.

MESH-PDMROF results imply a 70 percent difference between model simulated and the

Agriculture Canada $A_e$ delineations (Figure 3). This study doesn't consider the latter's

contributing area delineation an observation, and therefore there was no assumption taken that it

was more accurate than that of the modeled one. The observed contributing area differences

might be mainly due to the approaches employed to generate them. The Agriculture Canada

methodology employed topographic mapsheets and a suite of assumptions that was needed to

apply the method across the region. The La Salle watershed has very low relief outside the

northwestern portion, which is hummocky and populated by numerous depressions (Figure 1).

This is likely why the original $A_e$ delineation is so high (1645 km$^2$ or a $A_{ef}$ of 0.9).

The bias between simulated and observed $Q_{max}$ (Figure 3 and Table 2) indicates there is some

uncertainty in simulated estimates of the frequency with which areas contribute to $Q_{max}$. The

streamflow and contributing area estimated from MESH-PDMROF are only as good as the

forcing data and model structure allow. For instance, there is an assumption in MESH-

PDMROF of no surface-subsurface linkages, but these processes are known to be important for



maintaining surface stream connections and contributing areas (Brannen et al., 2015). The

model may compensate for this absent process by adjusting linear reservoir parameters K1ROW

and K2ROW (Table 1) during the calibration phase. This would affect estimates of contributing

area. To account for this uncertainty, frequency analysis was not only performed on simulated

values of $A_c$, but bias and error corrected values. Both corrections assumed that differences in

$Q_{max}$ estimates were proportionate to differences in $A_c$. Bias was addressed by adjusting all

values by 18% as noted above, while error was addressed by adjusting each annual value by the

fractional difference between simulated and observed $Q_{max}$. These values were bounded and

represent the possible range of simulated $A_c$ shown in Figure 3. The range defined in Figure 3

implies there is non-linear behavior between contributing area and return period distinctly

different than that of observed or simulated streamflow. Simulated streamflow displays

consistent increases in magnitude from the smallest events to approximately the 10 year return

period. Contributing area increases at different rates; expanding rapidly with the most frequent

streamflows, less quickly through conditions associated with the mean annual flood, increasing

to half of $A_g$ perhaps with the 1:5 year flood, but then stabilizing and increasing slowly through

the 1:10 flood. The notable difference in the Agriculture and Agri-Food Canada estimate of $A_{ef}$

(0.9) and simulated $A_{ef}$ that may range from 0.2 – 0.35 remains, even when using a bias or error

corrections based on simulation and observation difference.

Gray (1970) using data from Durrant and Blackwell (1959) provides a summary of the most

recent calculations of the relationship between gross drainage area and peak annual streamflow

for the Canadian Prairies. Among tributaries to the Red River (referred to as Region 9 by

Durrant and Blackwell (1959)), they identified a power-law relationship between $A_e$ and the



mean annual flood ($Q_{2.33}$) with an exponent value of 0.65 (Table 3). However, documented

trends in streamflow in this region from the second half of the 20[th] century (Rasmussen, 2016;

Burn and Whitfield, 2016) may result in a change in the exponent since the original period of

record in Durrant and Blackwell (1959) of 1911 – 1956. Using as many of the same stations as

5    possible, as some have closed, an analysis for the period coincident with this study (2002-2014),

suggests the regional $Q_{2.33}$-$A_e$ power law exponent has declined to 0.5 (Table 3). Assuming that

the entire gross drainage area contributes runoff during extremely large floods, the regional $A_g$

and $Q_{100}$ relationship was determined and found to have a marginally steeper slope with an

exponent of 0.54. Applying the range of contributing area estimates from Figure 3 demonstrates

10   that the relationship between La Salle River $Q_{max}$ and $A_c$ can be bounded by curves with

exponents between 1.12 and 0.89 (Figure 5).

Table 3: The form of regional Q-A relationships for Red River tributaries, and the La Salle River.

| Watersheds | Q-A relationship | Period | a | b |
|---|---|---|---|---|
| Durrant and Blackwell "Region 9" | $Q_{2.33} - A_e$ | 1911-1956 | 0.27 | 0.65 |
| Durrant and Blackwell "Region 9" | $Q_{2.33} - A_e$ | 2002-2014 | 1.13 | 0.5 |
| Durrant and Blackwell "Region 9" | $Q_{100} - A_g$ | 2002-2014 | 3.1 | 0.54 |
| La Salle | $Q_{max} - A_c$ (lower bound) | 2002-2014 | 0.08 | 0.89 |
| La Salle | $Q_{max} - A_c$ (upper bound) | 2002-2014 | 0.14 | 1.12 |





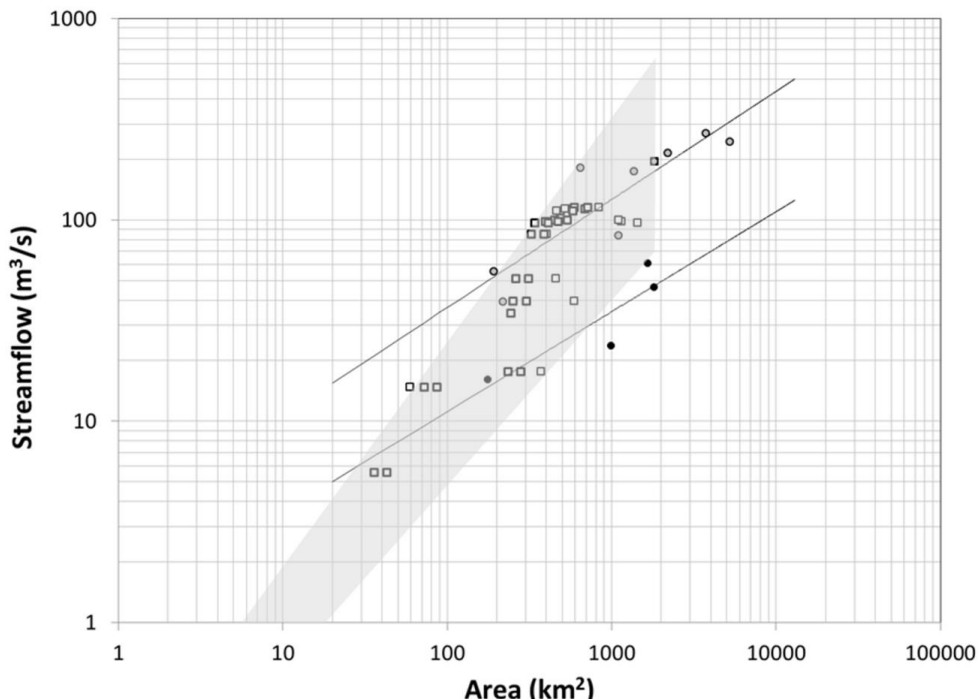

Figure 5: Area – streamflow relationships for tributaries of the lower Red River. Values from the watersheds denoted by the black dots were used derive the $Q_{2.33} - A_e$ relationship, while values from the watersheds represented by the grey dots were used to derive the $Q_{100} - A_g$ relationship, the two of which are the lower and upper lines, respectively. The white squares represent uncorrected and corrected simulated $Q_{max}$-$A_c$ values, which bounded by the grey transparent area, represent the best estimate of this relationship for the La Salle River.

## 5 Discussion

### 5.1 Contributing Area and Streamflow Behaviour

The results described above imply exponents of the power-law relationship between annual

maximum contributing area and streamflow tend to be greater than one. Using remotely sensed

estimates of contributing area, Mengistu and Spence (2016) derived exponents for three

headwater catchments at the St. Denis National Wildlife Area, in Saskatchewan, Canada, which





is a very hummocky portion of the Canadian Prairies. There, exponents ranged from 2.2 to 1.8, with smaller values associated with less depression storage within the watershed. The La Salle catchment has gentle relief and only ~10% of the basin is populated with widespread depressions (Figure 1) and exhibits an exponent range of 0.89-1.12. The pattern towards lower estimates of

power exponents in the La Salle from those of Mengistu and Spence (2016) support the results from Glaster (2007) who concluded that power-law relationships between area and streamflow converge near unity with increasing uniformity in hydrology of a basin.

The different scaling relationships among floods and contributing area in the La Salle watershed

are visualized in Figure 5. The envelope of MESH-PDMROF corrected and uncorrected estimates of contributing area demonstrates that in the La Salle basin, streamflow increases with contributing area at a faster rate than regional streamflow would with gross drainage area during the same magnitude flood. This characteristic implies that the suite of regional streamflow-area curves of different return periods is merely the product of the annual streamflow-contributing

area curves from individual basins (Figure 6). Forsaith (1949) plotted floods of different return periods against the gross drainage area of several Canadian prairie streams to determine if the scaling exponent would change with flood magnitude. It remained at a value of 0.5. However, the coefficient was found to change with return period following:

$$Q_{tr} = A_g{}^{0.5} \cdot 0.2842 \cdot tr^{0.444} \tag{6}$$

where tr denotes return period. However, this relationship applies the constant gross drainage area. Different exponents of the regional $Q_{100}$-$A_g$ and $Q_{2.33}$-$A_e$ curves (0.53 and 0.5) derived from the Agriculture Canada (1983) estimates suggest the scaling relationship may change with flood magnitude. In the absence of anything to suggest otherwise, this change in the scaling




relationship with flood magnitude can be assumed to be linear. This adjusts the relationship

between area, return period and streamflow to:

$$Q_{tr} = (0.0201 \cdot tr + 1.0883) \cdot A^{(0.0004 \cdot tr + .4956)} \tag{7}$$

or rearranged:

5  $$A = \left(\frac{Q_{tr}}{(0.0201 \cdot tr + 1.0883)}\right)^{1/(0.0004 \cdot tr + 0.4956)} \tag{8}$$

to provide a method to estimate $A_c$ using annual maximum streamflow of a specific return

period.    Once associated with acceptable certainty, regional data could be used to develop

functions useful for determining the contributing area associated with any regional annual

maximum streamflow.

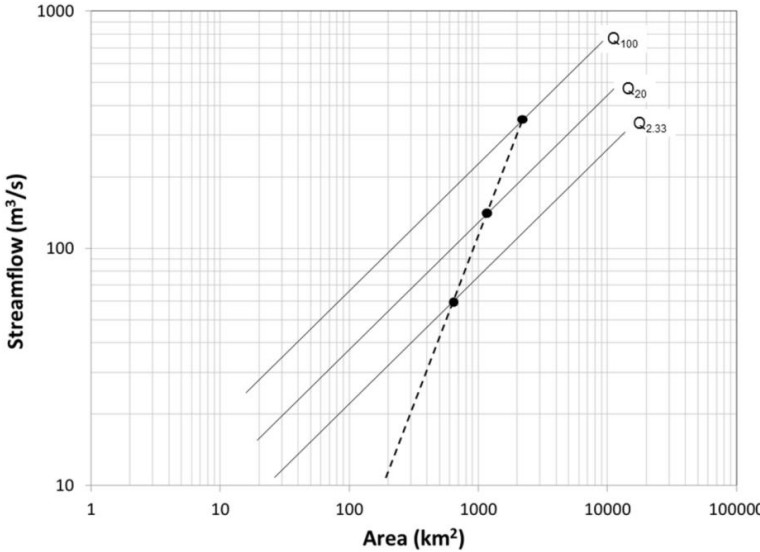

Figure 6: Conceptual curves of the relationship between an individual catchment $Q_{max}$-$A_c$
function, and how it influences the shape of the regional area – streamflow curves.



The coefficients and exponents in Eq. 8 are distinct to southern Manitoba, but the form of the equation should be applicable elsewhere, and could be used as a hypothesis to determine its ubiquity. There are three current primary uncertainties associated with Eq. 8 that prevent its practical use. The mean annual flood curve is based on the assumption that the original $Q_{2.33}$-$A_e$

estimates are well-constrained, which the MESH-PDMROF are not, at least in the La Salle River basin. Second, it is assumed the gross drainage area equates to contributing area during the 1:100 year flood, which has not been evaluated with observations. Third, the inherent scatter in behavior among watersheds even within regions considered homogenous can be large enough to create poor results. The curves in Figure 6 contain enough uncertainty that they should be

considered conceptual in nature, and not be used for practical purposes.

The technology exists to generate the necessary data to constrain the curves. Remote sensing products from satellite and drone platforms have proven useful in estimating saturated and inundated areas at the required resolutions (Phillips et al., 2011; Spence and Mengistu, 2015).

Alternate models that employ digital elevation models and simple water budgets have proven useful for evaluating contributing area dynamics (Shook and Pomeroy, 2011). These methodologies require robust field observations for calibration and validation. A field program that includes contributing area mapping with associated hydrometric measurements in a selection of sites with representative landscapes would help to develop estimates useful for constraining

regional Q-A curve estimates. Furthermore, it would assist in developing relationships among topographic and topological parameters useful for learning what controls the coefficient and exponent values of the Q-A relationship. These could be extrapolated across regions to evaluate impacts of land management practices that alter the nature of contributing area dynamics.



### 5. 2 Implications for Nutrient Management

Because of the extreme nature of contributing area dynamics in the Canadian prairie region, in particular, an investigation of what controls the Q-A relationship would be exceptionally

important. This could inform practices to mitigate lake eutrophication apparent in several prairie watersheds.  Runoff from naturally nutrient rich prairie soil landscapes and predominantly agricultural lands has been blamed for carrying sediment and nutrients to Lake Winnipeg, and enriching it enough to cause excessive primary productivity (eutrophication) in the lake (Schindler et al., 2012; McCullough et al., 2012). Investigation of recurrence of contributing area

in such systems can inform how often the sources and pathways of nutrient loaded runoff connects and disconnects to the stream network, and help to recommend where best to direct sound strategies for effective implementation of drainage management activities geared towards reduction of pollutant loading to the lake.

Recent years have been wetter than normal in southern Manitoba, with the frequency of large floods increasing since the late 1990's (Burn and Whitfield, 2016; Rasmussen, 2016).  The Red River as measured at Emerson experienced a 1957 – 2016 average $Q_{max}$ of 975 m$^3$/s.  The number of above average annual floods in the first half of this 60 year period was 12; increasing to 17 since 1987.  The non-linear shape of the Q-A$_c$ relationship (Figure 5) implies that this trend

towards a higher streamflow regime has likely been associated with a significant expansion in the area capable of acting as a source for nutrients in recent years.  Average $Q_{max}$ from 1957 – 1986 was 835 m$^3$/s, increasing 33% to 1120 m$^3$/s between 1987 and 2016.  Transferring $Q_{max}$-A$_c$ relationships from the La Salle (Table 3) implies this could equate to 13000 more square





kilometers, an increase of 39%, and 12% more of the Red River watershed now regularly acting

as a source for solutes and nutrients downstream. As alluded to above, the uncertainty in these

relationships means that these estimates should be treated with caution.

However, with these current tools it remains unclear exactly where this expansion has taken

place, and in turn, where landscape management efforts to reduce nutrient loading to higher

order streams and lakes should be focused. The science is still evolving in regards to evaluating

predominant hydrological and hydrogeological processes in a prairie landscape with an

important wetland complex and how these processes ebb and flow over longer drought and

pluvial periods (Hayashi et al., 2016). This creates difficulties in informing what could be the

optimal wetland complex for flood protection, aquatic ecosystem resilience, nutrient

management and rural water supply. MESH-PDMROF estimates a catchment-wide estimate of

contributing area fraction, unlike the distributed models of Smith et al., (2013) and Nippgen et al.

(2015). A next generation version of MESH that is capable of simulating the spatial distribution

of contributing area would assist in addressing these types of important land and water

management questions.

## 6 Conclusion

Simulations of streamflow and contributing area in the La Salle River basin in southern

Manitoba demonstrate that the frequency distributions of contributing area and floods may not be

the same. This is due to a non-linear power scaling function between flood magnitude and

contributing area. The results presented here suggest that individual catchment functions shape

regional area-flood-return period relationships. Applying the modelled estimate of the form of





the power-law function for this region suggest that a 33% increase in mean annual maximum
streamflow in the larger Red River into which the La Salle River flows, could be associated with
an increase in mean annual maximum contributing area of up to 39%. Uncertainty bounds in the
model results mean these estimates are preliminary and error was too broad to produce curves

well constrained enough for practical purposes. Contributing areas which are sources of solutes
and nutrients can be activated in a non-linear manner. The methods introduced here provide a
means to quantitatively assess this activation. It is recommended that future research build upon
past field and modelling experiments to determine how the distribution of landscape features
such as depressions and hillslopes control the contributing area regime and incorporate this into

model parameterization schemes and algorithms. This would help inform what controls the
expansion and contraction of contributing areas in this and other landscapes. Furthermore, this
would permit spatially distributed estimates of contributing area behaviour and aid in more
informed decisions of how to manage landscape features to meet societal goals of improved
water quality and flood protection.

**Acknowledgements**

The authors wish to thank Al Pietroniro, Bruce Davison, Issac Wong and Ram Yerubandi of
Environment and Climate Change Canada and Muluneh Mekonnen of Alberta Environment for
their assistance during this project. This research was funded by Environment and Climate

Change Canada's Lake Winnipeg Basin Initiative. Samson G. Mengistu was supported by the
National Science and Engineering Research Council (NSERC) Government Laboratory Visiting
Fellowship program.



**COMPETING INTERESTS:** The authors declare that they have no conflict of interest.



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
