# Peer review of "On the Relationship Between Flood and Contributing Area"

_Hydrology and Earth System Sciences, 2017_

## Referee Comment (RC1) · Anonymous Referee #1 · 8 Aug 2017

I have completed my review of the paper "On the relationship between flood and contributing area" by C. Spence and S.G. Mengistu, submitted to HESS. This paper attempts to investigate the relationship between streamflow ($Q$) and contributing area ($A\_c$) using a mix of modelling and very limited data. The goals of the paper, as indicated by the authors are (1) "to test the hypothesis that the relationship between a catchment's floods and contributing area is a power function" and (2) to compare contributing area and flood frequency distributions. While the context and motivation for the work is sound and very well expressed (the introduction was a very enjoyable read), the execution of the methodology and the interpretation of results were plagued by a number of critical problems. I have three primary concerns about this paper which I believe make it unpublishable, even with major revisions. These are reflected in my

major comments #1, 2, and 3 below.

Major Comments (1) One of the critical issues in this paper is that the authors used a model which implicitly assumes a functional form between contributing area and discharge from the landscape. The PDMROF model used here, based upon the Probability Distributed model of Moore (1986), generates runoff as the water applied to the percentage of water stores which are filled to capacity, effectively an approximation of saturation excess on a heterogeneous landscape. While the Q-A relation individual flood events will be modified by the available runoff and be somewhat sensitive to the shape of the pareto distribution and the current soil moisture deficit, for a given time step, the volume released will be roughly linearly proportional to the percentage of filled water stores, as is clear from figure 5 – the PDMROF results fall within a b of 0.89 and 1.12: a linear model (b=1) is a very reasonable fit to the data. Since they are using the percentage of filled water stores as a proxy for contributing area, and not reporting whether or not they are using mean hourly flow/flow at the end of the time step or which version of contributing area (start of time step, mean, or end of time step), the scatter in this modeled data around the 1:1 line could easily be a temporal discretization artefact.

If the storage distribution is treated as a pareto (i.e., power law) distribution with a scale parameter (i.e., minimum storage) of 0, this will necessarily lead to an outflow-contributing area relationship which will be something very close to a simple power law function. This makes the hypothesis test (that the catchment flood and contributing area can be fit with a power function) straightforward, as the modeled results will certainly echo this – *this assumption is built into the model used to test the hypothesis*, which is the key issue. One cannot use a model to test a hypothesis which is itself built into the model.

Because of the built-in assumptions of PDMROF, I am doubtful that one can use a model such as the one used here (MESH) to test the primary hypothesis of this paper that the relationship between a catchment's floods and contributing area is a power function. A more appropriate approach would be to use a model which simulates all of

the connections and overflow thresholds in the flow network while making no assumptions about the contributing area/outflow relationship.

(2) The estimated scaling relationship to generate the Q_tr-A_c relationship of equation 8 is generated using 2 data points and untested on verification data, yet the authors report coefficients to 4 significant digits. While it is clear the authors recognize many of the significant issues with using this equation (they catalogue many of them), I don't believe there is any justification to include it in the paper at all. The relationship is purely hypothetical and not supported through testing or comparison to additional data. The authors cite a list of reasons (in this case, "uncertainties") which describe why the expression is flawed, with insufficient evidence to supports its use.

(3) This model is of insufficient quality (as indicated in figure 3) to support the stated hypotheses. "Cleaning this data" with bias/error corrections does not fix the problem.

(4) The comparison of the model results with the (literally one data point) Agriculture Canada Ae delineations is weak. The authors acknowledge this: "The observed contributing area differences might be mainly due to the approaches employed to generate them". I agree with this statement. However, they move forward with this comparison as if there were more to it, introducing the bias and error corrections (which were not explained with sufficient detail to understand or replicate) to weakly assess model uncertainty.

Minor Points (given my recommendation, a non-comprehensive list) (1) The discussion of the history of MESH on pg 8 could be relegated to a simple citation of Pietroniro et al. (2007)

(2) Contrasting the GRU approach and PDMROF on page 9 and suggesting that PDMROF is an improvement over the GRU approach makes little sense; one is a discretization approach, one is a means of representing the flow-storage relation (i.e., the hydrologic process description). PDMROF could easily be applied in a GRU context, and actually is within the model used in this study. Rather, PDMROF is a replacement for

the runoff representation of WATROF.

(3) pg 11 ln 18- grid resolution of 10km or 10km^2? The former implies only ~20 grid cells were used.

(4) pg 12- no need to define NSE in such detail.

(5) pg 12 – which Ostrich calibration algorithm was used?

(6) eqn 2 – this test statistic should be cited.

(7) pg 14 – this entire discussion of probabilities and plotting position could be relegated to a citation.

(8) pg 15 – a mean annual flood error of 100%

(9) It would be useful (for clarity) to split figure 3 into an Area distribution plot and a flood frequency distribution plot.

(10) The concepts in the nutrient management discussion seem only peripherally linked to the main objectives of the paper.

---

## Referee Comment (RC2) · Anonymous Referee #2 · 14 Aug 2017

The Spence and Mengitsu manuscript focuses on the characterization of the flood and contributing area relationship for an agricultural catchment in Canada. The flood data is generated using a semi-distributed model (MESH-PDMROF) and the contributing area fraction is generated from the same model with the method from their previous paper (Mengitsu and Spence, 2016). Although this manuscript is well written, I too have concerns regarding the methods of this paper, but will try to focus on the aspects that were not covered by Reviewer 1.

(1) First of all, I disagree with the authors that the MESH-PDMROF model deployed here provides reasonably well simulations. Figure 2 compares the observed and simulated hydrographs, and the model underestimates most of the hydrograph event peaks (by a large margin!) in the 12 year period. This is especially important because the

study's main focus is on floods. As described by the authors, MESH-PDMROF has a SWAT-like structure. However, to my knowledge, application of SWAT in agriculture dominated catchments requires additional process modules (e.g., a tile drain module). None of that seems to have been implemented here with MESH-PDMROF. Therefore, my suspicion is that the model (as implemented in current form) might be missing some key processes that are important to this catchment. This also brings into question the validity of the contributing area calculations.

(2) The error and bias correction of the data further mangles the situation because of the assumption that the differences in Qmax between the observed and simulated data are proportionate to differences in Ac. This is a reasonable assumption in itself, but the underlying premise is that any deficiencies in the model can be fixed through a bias correction (which I don't think is appropriate). Therefore, it is unclear what to make of the Q-A relationships and exponent values presented later in the Results.

---

## Author Comment (AC1) · 20 Sep 2017

With the reviewers' comments not being supportive of publication of this manuscript in its current form, we feel it is doubtful that it will be allowed to continue in the HESS peer review process. So, we are going to take this opportunity to follow the spirit of HESS-D and provide rebuttals and agree where appropriate, but also challenge the reviewers about their assumptions of the challenges of modelling Canadian Prairie streams, and expectations of existing model structures in this environment. It is clear as authors we need to reframe the discussion about what has been learned by this research. We would like to thank the reviewers for their input and hope they are open to continue this debate.

[Figure]

Some of both reviewers' major comments overlap into two major concerns. First, the ability of MESH-PDMROF to produce workable results that can be used to answer the research questions. Second, the use of bias and error correction to improve the context of the mediocre results. These are valid concerns; ones we certainly struggled with. That said, this application of MESH-PDMROF performs at the higher end of documented applications of similar models used in this region. Yang et al. (2011) using SWAT achieved an NSE of 0.2 in Broughton's Creek. Shrestha et al. (2011), using SWAT, had validation NSE's in the Morris basin that ranged from 0.65-0.19 depending on the precipitation dataset. Shrestha et al. (2012) also applied SWAT in the Upper Assiniboine, with validation NSE's of 0.65. Can either reviewer provide an example of a semi-distributed model that includes all the hydrological processes relevant to the Canadian Prairie that could be reasonably applied to a 2000 km2 watershed? SWAT does not include the snow and frozen soil processes that MESH does. CAN-SWAT is better, but it suffers from having a GRU type structure that assumes all upslope runoff can reach the stream, which is violated almost everywhere in the Canadian Prairie. MESH-PDMROF does not carry this assumption, which is crucial to the estimation of contributing area. As pointed out by reviewer #2, MESH does not have a tile drain function. CRHM doesn't have the proper stream routing. Frankly, we were surprised we did this well. Providing uncertainty bounds, which was our goal with the corrections, was prudent. It does not fix the problem of inadequate model structure, but consider that MESH-PDMROF is still a state of the art tool, for all its faults. MESH-PDMROF is the only existing semi-distributed model proven to reasonably estimate contributing area in this environment. It was not the objective of this research to improve MESH-PDMROF. Others are doing this because the lack of a robust model for this region seriously hampers the generation of good information necessary to properly inform on-farm land management decisions. Our objective was to assess the nature of streamflow – contributing area relationships in this landscape with the best available tools. In regards to the correction process, we do not claim that these are to fix model deficiencies. The correction process is to provide some uncertainty bounds around our estimates. Do

the reviewers have suggestions for improving the uncertainty assessment and estimating these bounds? These would be very welcome for any re-submitted manuscript to this journal or another.

Reviewer #1 had several other major and minor comments. We address these in turn. Major comment #1 had a couple sub-components to it.

1a) We are unsure to which 1:1 line the reviewer is referring, as there was no figure showing scatter. Perhaps the reviewer was referring to the range of b from 0.89 to 1.12. In response, the model runs always used the end of the time step for all terms. In any resubmitted version of the paper to this journal or another we will be explicit on this matter.

1b) The reviewer's point about using a model with a Pareto distribution as a scale parameter is fair. This hypothesis will be removed from any future submission. We still believe that an interesting contribution to the literature would be one that addresses the remaining research question; Are regional flood frequency curves a construct of individual catchment contributing area-flood curves? I guess the primary debate here is, can we determine this in southern Manitoba with MESH-PDMROF?

2) In regards to the scaling relationship between return period, contributing area and streamflow, of course equations 7 and 8 are purely hypothetical and untested. The reviewer probably realizes that the technology to measure contributing area over a $\sim$2000 km2 watershed is in its infancy and that the means to provide data for testing barely exists, never mind the lengthy contributing area time series data he/she implies someone might have. This is a huge knowledge gap in this region, and across the world, with massive implications for understanding how watersheds filter inputs and release water, sediment and solutes. Hence the form of Figure 6, which provides conceptual curves of the form of equations 7 and 8. Even with the wide uncertainty in the model predictions, the slopes of these curves are constrained enough to imply that regional flood frequency curves are a construct of individual catchment contributing area-flood

curves. One of the key outcomes of the research is the recommendation that theoretically one could calculate annual maximum contributing area with an annual flood estimate with an equation of this form. Equations 7 and 8 are really only rearranged versions of equation 6 with the assumption that the change in the contributing area - flood scaling relationship with flood magnitude is linear. Perhaps a better approach would have been to keep the derivation of these equations purely mathematical, and propose equations 7 and 8 as hypotheses for future testing. Perhaps we and the reviewer can agree that including numbers insinuates confidence in these equations for practical use. It would seem exceptional confidence with that many significant digits. Our mistake.

3) To not include the Agriculture Canada estimate would have been a significant omission. It needs to be included because those estimates are used commonly across the Canadian Prairie to inform analysis of streamflow response, flood frequency, and nutrient transport, among others. The frequency distribution of the modelled data implies the modelled mean annual flood was half that estimated with Water Survey of Canada data. There are two simple ways to address any associated error in the modelled contributing area. Double the modelled contributing area fraction estimate or use the modelled contributing area fraction estimate associated with the 'observed' mean annual flood amount. Both of these are $\sim$0.4. This is still a little more than half the Agriculture Canada estimate. Does the large difference not intrigue the reviewer? This, at least, suggests our community does not have tools to constrain regional estimates of contributing area (as alluded to above). At most, it introduces doubt into all the Agriculture Canada estimates across the Canadian Prairie and suggests they need to be re-evaluated with new modelling methodologies and a robust observation program. The uncertainty needs addressing, which is critical for sound assessments of climate and landscape management impacts on floods and nutrient export in this region. The modelling results should not be dismissed but a call to action that shows how little we know and can predict contributing area dynamics. Non-parameterized versions of equations 7 and 8 could be used as a hypothesis framework for this research.

Reviewer #1's minor points:

1) I have always found it interesting what different reviewers claim important. If we had left this content out, there was an equal chance that another reviewer would have asked to have it in.

2) We agree.

3) 10 km2

4) See response to minor point (1) above.

5) We used the DDS (Dynamically-Dimentioned-Search) algorithm.

6) Will do.

7) See response to minor point (1) above.

8) Yes.

9) We like them together because it displays the differences in return periods very well.

10) Thank you for noting this. Given our rant above, it may be clear that the issues of contributing area and nutrient management are inextricably linked in this region (i.e., wetland drainage). More content on this matter is clearly needed early in the paper to ensure readers get the information needed to understand the Canadian Prairie context.